# Theaflavin Ameliorates *Streptococcus suis*-Induced Infection In Vitro and In Vivo

**DOI:** 10.3390/ijms24087442

**Published:** 2023-04-18

**Authors:** Ting Gao, Yiqing Tan, Yanjun Wang, Fangyan Yuan, Zewen Liu, Keli Yang, Wei Liu, Rui Guo, Chang Li, Yongxiang Tian, Danna Zhou

**Affiliations:** 1Hubei Provincial Key Laboratory of Animal Pathogenic Microbiology, Key Laboratory of Prevention and Control Agents for Animal Bacteriosis, Institute of Animal Husbandry and Veterinary, Hubei Academy of Agricultural Sciences, Wuhan 430064, China; gaotingyefeiyeziyu@163.com (T.G.); tyq0517@foxmail.com (Y.T.); wangyanjun202202@163.com (Y.W.); liuzwen2004@hbaas.com (Z.L.); keliy6@hbaas.com (K.Y.); liuwei@hbaas.com (W.L.); guorui@hbaas.com (R.G.); lichang1113@hbaas.com (C.L.); 2State Key Laboratory of Agricultural Microbiology, College of Veterinary Medicine, Huazhong University, Cooperative Innovation Center of Sustainable Pig Production, Wuhan 430070, China

**Keywords:** *Streptococcus suis*, theaflavin, suilysin, molecular docking, hemolytic activity

## Abstract

*Streptococcus suis* (*S. suis*) is one of the most important zoonotic pathogens that threaten the lives of pigs and humans. Even worse, the increasingly severe antimicrobial resistance in *S. suis* is becoming a global issue. Therefore, there is an urgent need to discover novel antibacterial alternatives for the treatment of *S. suis* infection. In this study, we investigated theaflavin (TF1), a benzoaphenone compound extracted from black tea, as a potential phytochemical compound against *S. suis*. TF1 at MIC showed significant inhibitory effects on *S. suis* growth, hemolytic activity, and biofilm formation, and caused damage to *S. suis* cells in vitro. TF1 had no cytotoxicity and decreased adherent activity of *S. suis* to the epithelial cell Nptr. Furthermore, TF1 not only improved the survival rate of *S. suis*-infected mice but also reduced the bacterial load and the production of IL-6 and TNF-α. A hemolysis test revealed the direct interaction between TF1 and Sly, while molecular docking showed TF1 had a good binding activity with the Glu198, Lys190, Asp111, and Ser374 of Sly. Moreover, virulence-related genes were downregulated in the TF1-treated group. Collectively, our findings suggested that TF1 can be used as a potential inhibitor for treating *S. suis* infection in view of its antibacterial and antihemolytic activity.

## 1. Introduction

The tea plant *Camellia sinensis* (L.) O. Kuntze is a source of commercially grown tea, which is the second most popular beverage, after water, in the world. There are three major types of tea widely consumed based on the degree of fermentation, black tea, green tea, and oolong tea [1]. Tea consumption has also been connected with health benefits on account of the health-promoting ingredients, including catechins, theaflavins (TFs), flavonoids, theanine, and so on [2]. Many studies have revealed that the above bioactive compounds have excellent anticancer, antioxidant, antiobesity, anti-inflammatory, and antimicrobial properties [3,4].

Black tea is the most consumed tea that accounts for 78% of worldwide tea consumption and is popular in the United States, Europe, Africa, and India [5]. Traditionally, the manufacturing process of black tea involves four steps, withering, rolling, fermentation, and drying [6], among which fermentation is the most important step. After these processes, numerous bioactive compounds are formed in black tea, such as amino acids, alkaloids, catechins, theaflavins, and thearubigins. Theaflavins are the main oxidation products of epicatechin, epicatechin-3-gallate, epigallocatechin and epigallocatechin-3-gallate in green tea and aggregate during the fermentation step [7]. Usually, there are four major theaflavins in black tea, theaflavin (TF1), theaflavin-3-O-gallate (TF2a), theaflavin-3′-O-gallate (TF2b), and theaflavin-3,3′-O,O-digallate (TF3). Yet, despite TFs having many pharmacological activities, there are only a few researches concerning the bactericidal effect of TFs. Until now, TFs were reported to inhibit *Streptococcus mutans*, *Bacillus coagulans*, *Acinetobacter baumannii*, and *Bacillus cereus*. Some bacteria are sensitive to TF1, other bacteria are not. No study has revealed whether TFs had antibacterial activity against zoonotic *Streptococcus suis* yet [8,9,10,11].

*Streptococcus suis* (*S. suis*) is a major cause of pneumonia, arthritis, and meningitis in the pig industry, and an emerging pathogen in humans. *S. suis* infection has been reported in humans in more than 30 countries or regions of the world [12]. In China, *S. suis* has previously received much attention on account of two large outbreaks of human infection in 1998 and 2005, resulting in at least 237 infections and 53 deaths [13,14]. There are 33 serotypes of *S. suis* based on a capsular polysaccharide, among which serotype two (SS2) is considered to be the most virulent and prevalent strain isolated from diseased pigs and human beings [15,16]. The traditional treatment of *S. suis* infection depends on antibiotics, just like other bacterial infections. A vaccine against *S. suis* could not provide effective protection for immunised pigs and there is no *S. suis* vaccine for humans. For patients, *S. suis* infections are usually administrated with penicillin G, accompanied by one or more other antibiotics including ceftriaxone, gentamicin, chloramphenicol, and ampicillin [13]. For pigs, *S. suis* infections are often treated with fluoroquinolones, aminoglycosides, and β-lactams in the swine industry [17]. In recent years, with the abusive use of antibiotics, an increase in multidrug-resistant bacterial strains has led to the poor performance of traditional antibiotic therapies [18,19]. Among 506 *S. suis* isolated from pig farms in the Czech Republic from 2018 to 2022, a high frequency of resistance was found in clindamycin, tilmycosin, tulathromycin, and tetracycline [20]. In the period from 2017 to 2019, a total of 314 nasal swab samples were obtained from clinically healthy pigs in China with 34.08% isolation of *S. suis*, and a high level of resistance to clindamycin, tetracycline, clarithromycin, and erythromycin was observed [21]. Based on the above epidemiological investigation, alternative antibacterial agents with high efficiency and no drug resistance are urgently needed.

In previous studies, some natural compounds have been reported to inhibit *S. suis*, including green tea polyphenols, baicalein, elipticine hydrochloride, amentoflavone, fisetin, and myricetin, which can weaken *S. suis* pathogenicity by inhibiting the hemolytic activity of suilysin (Sly) [22,23,24,25]. Sly is a secreted protein that can create pores in the target host cell membrane and contributes to the successful colonization of host cells and immune escape of *S. suis*. Many research findings revealed that *S. suis* strains with high levels of Sly production are more likely to cause high mortality in infected mice than nonvirulent strains; in other words, indicating that the pathogenicity of *S. suis* can be decreased by lowering the production of Sly. Therefore, Sly plays a vital role in the stage of *S. suis* infection, and Sly may be a promising new target for the treatment of *S. suis* infection. However, the potential application of TF1 in veterinary clinical treatment and prevention has been poorly investigated, and the study of the interaction between TF1 and Sly is still in the blank.

In this study, we investigated the antibacterial effect of theaflavin (TF1) against SS2 in vitro and in vivo. Our data demonstrated that TF1 not only inhibited SS2 growth but also caused damage to bacteria, exhibited strong antihemolysin activity and reduced biofilm formation. In particular, animal experiments indicated that TF1 could impair the ability of *S. suis* to adhere to NPTr epithelial cells, improve the survival rate of SS2-infected mice, and relieve the clinical symptoms of SS2-infected mice. Overall, our results indicated that TF1 could be a potential phytochemical compound for treating *S. suis* infection.

## 2. Results

### 2.1. Inhibition of S. suis Growth by TF1

TF1 extracted from black tea was quantitatively identified by HPLC, and the purity of the TF1 was 99.16%. The chemical structure of TF1 was shown in Figure 1A. The MICs of TF1 against Escherichia coli were ranging from 2048 to 4096 μg/mL and the MBCs were 4096 μg/mL. The MICs of TF1 against *Streptococcus suis* were 512 μg/mL and the MBCs were 2048 μg/mL (Table 1). *Streptococcus suis* was more sensitive to TF1 than *Escherichia coli*. Then, the most virulent *Streptococcus suis* strain SC19 was chosen for the following experiments. The time-killing curves demonstrated that TF1 exerted effective killing effects on SC19. The growth of SC19 was inhibited by TF1 at MIC, and after coincubation at MBC for 4 h, the bacteria were thoroughly killed (Figure 1B,C), indicating that TF1 effectively killed SC19 in a dose-dependent manner.

### 2.2. TF1 Causes Damage to the Internal and External Structure of S. suis

The effect of TF1 on the morphology of SC19 was directly visualized by a transmission electron microscope (TEM) and a scanning electron microscope (SEM). After treatment with TF1 at MIC, SC19 was severely damaged, the cell wall and cell membrane showed breakage, and the entire bacterium was only an empty shell left owing to the leakage of cytoplasmic content (Figure 2). Moreover, most cells were seriously shrunken and presented an irregular shape, on account of the indistinct cell wall and collapsed cell membrane.

### 2.3. TF1 Inhibits Hemolytic Activity and Biofilm Information of S. suis

The hemolytic activity of SC19 was markedly reduced by TF1 at MIC (*p* < 0.001), when the concentration of TF1 increased to MBC, the hemolytic activity of SC19 was threefold less than that for the untreated SC19 (*p* < 0.001) (Figure 3A,B). In addition, further study was performed to find whether the decreased hemolytic activity of SC19 was due to the inhibition of TF1 on bacterial growth or due to the direct inhibition of TF1 against Sly. As expected, the hemolytic activity of Sly protein in the culture supernatant of SC19 was also damaged by TF1 (Figure 3C,D), indicating the direct interaction of TF1 and Sly. However, the declined degree of hemolytic activity of the SC19 supernatant was less than that of the SC19 culture.

As shown in Figure 3E,F, two different doses of TF1 were tested against SC19 biofilm formation. At MBC of TF1, there was significant inhibition (*p* < 0.0001) of S. suis biofilm formation, but it had no effect at MIC (*p* > 0.05), suggesting that 2048 µg/mL was more effective than 512 µg/mL at inhibiting biofilm formation.

### 2.4. TF1 Decreased Adherent Activity of S. suis to Epithelial Cell Nptr

Lactate dehydrogenase (LDH) release measurements were performed to determine whether TF1 could be cytotoxic to the epithelial cell Nptr. Figure 4A showed that TF1 with different concentrations did not injure Nptr at all. The adhesion assay revealed that TF1 also decreased the adherent activity of *S. suis* at 1/4 MIC (*p* < 0.05), when the concentration of TF1 rose to 1/2 MIC, the binding rates to the Nptr cells of SC19 were threefold less than that of the untreated group (*p* < 0.001) (Figure 4B).

### 2.5. TF1 Protects Mice against SC19 and Decreased the Colonization Ability of SC19

The protective effect of TF1 on infected mice was assessed based on the survival rate and colonization. All mice infected with SC19 that developed severe clinical symptoms, such as ruffled fur and septicemia, died within 2 dpi, and the survival rate was zero. Conversely, the survival rate of TF1-treated infected mice increased to 40% compared to that of the untreated infected mice, and the clinical symptoms were also mild (Figure 5A). Next, the effect of TF1 on the bacterial load of SC19 in different tissues of mice was investigated. The bacterial number of SC19 was much higher at 0.5, 1, and 1.5 dpi than that of the TF1-treated group in the brain, lung, and spleen (Figure 5B–D). Particularly, the bacteria in the mouse spleen were most cleared at 1.5 dpi. In addition, the expression levels of IL-6 and TNF-α in the blood of TF1-treated mice at 0.5 dpi were significantly decreased compared with that of the untreated group, however, there were no differences at 1 and 1.5 dpi (Figure 4C,D).

### 2.6. TF1 Binding Site on Sly Revealed by Molecular Docking

The TF1 binding site on Sly was identified by the molecular docking-based calculation. There were 32 binding pockets for Sly (Table 2), and the highest-scoring pocket was chosen for docking. As shown in Figure 6A,B, the 3D structure indicated the action mode of TF1 and Sly. The blind docking results revealed that a total of five docking phases were generated (Table 3), and the optimal (or minimal) energy was −6.3573 kcal/mol. More specifically, the two-dimensional configuration map in Figure 6C clearly exhibited that TF1 formed hydrogen bonds with Glu198, Lys190, Asp111, and Ser374, respectively. Hydrogen bonding was the leading drive for interaction between TF1 and Sly.

### 2.7. TF1 Inhibits the Expression of Virulence-Related Genes of SC19

To determine the effect of TF1 on virulence-related genes, a qRT-PCR was performed to evaluate the expression of Sly, EF, OTC, Gor, and GAPDH in SC19. Transcriptional analysis revealed that Sly, EF, OTC, Gor, and GAPDH were downregulated to a higher extent in TF1-treated SC19 than that which were in untreated SC19 (Figure 6D). This result was consistent with the above experimental data.

## 3. Discussion

With the overuse of antibiotics, the drug resistance of pathogenic bacteria is increasing year by year. Phytochemicals produce activity in many aspects, including antibacterial, anti-inflammatory, and antioxidative [26]. They consist of different sorts of compounds based on the chemical structures of the skeleton [27,28]. In this study, we demonstrated that TF1 extracted from black tea exhibits antimicrobial activity towards *S. suis* which is a zoonotic pathogen and causes severe economic losses. We observed that 512 μg/mL of TF1 inhibited the growth of SC19, whereas 2048 μg/mL of TF1 exerted bactericidal activity against SC19. After 4 h of coincubation with TF1 at a concentration of 2048 μg/mL, the bacteria were thoroughly eliminated. Although TF1 has been reported to inhibit many pathogenic bacteria [29], this is the first study to show that TF1 can ameliorate *Streptococcus suis*-induced infections in vitro and in vivo.

Virulence factors are promising antiinfection targets and avenues for vaccine development against pathogenic bacteria [30]. Sly is an important virulence factor that can create pores in the target cellular membrane, resulting in high levels of inflammasomes and meningitis caused by *S. suis* infection [31,32]. Therefore, the antihemolytic strategy is a new selection in target discovery. Of course, Sly is an effective target for treating *S. suis*-induced diseases [33]. In this study, TF1 not only significantly reduced the hemolytic activity of SC19, but also inhibited Sly protein in the culture supernatant of SC19. Subsequently, molecular docking suggested that there was one potential binding pocket in the protein Sly, and TF1 interacted with Sly through hydrogen bonds with four amino acids. It is obvious from these interactions that Glu198, Lys190, Asp111, and Ser374 are essential for the hemolytic activity of Sly. Therefore, TF1 was an effective inhibitor for Sly.

Animal and cell infection experiments were consistent with the above result. TF1 treatment significantly decreased the adhesion ability of SC19, causing alleviated cell damage, a high survival rate of infected mice and reduced production of the inflammatory cytokines TNF-α and IL-6. Although the contribution of Sly in the bacterial adherence and invasion of host cells is still not fully understood, Sly-positive *S. suis*-like B831, P1/7, and H11/1 were able to invade into HEp-2 cells at a higher percentage compared to the Sly-negative TD10, O891, and DH5 bacterial strains [34]. In other words, adherence and invasion abilities were positively regulated by Sly. Since TF1 inhibited the expression and activity of Sly, it was easy to explain why the adherence activity of TF1-treated SC19 was decreased. TF1 had anti-inflammatory activity in various models, such as cell, muscle, and brain [35,36,37]. A previous study revealed that TF1 reduced the expression of inflammatory factors (IL-1β, IL-6, and TNF-α) by regulating the TLR4/MyD88/NF-κB signalling pathway to alleviate muscle inflammation [36]. Lun et al. demonstrated that Sly triggered TNF-α production in human monocytes and IL-6 production in pig PAM cells and monocytes [38]. In our study, after treatment with TF1, TNF-α and IL-6 production in the blood of mice infected with SC19 at 0.5 dpi were decreased compared to the untreated group, though there was no significant difference at 1 dpi and 1.5 dpi. This result may be related to individual differences in the mice.

In summary, our data suggest that TF1 can provide a novel therapeutic method for *S. suis* infection due to its antibacterial and antihemolysin activities, which indicates that TF1 is a promising candidate for treating *S. suis* infections at the time when we are trying to reduce the use of antibiotics in pig medicine.

## 4. Materials and Methods

### 4.1. Bacterial Strains, Growth Conditions and TF1

The *S. suis* strain SC19 [39] was routinely cultured in tryptic soy broth (TSB; BD, Rockville, MD, USA) liquid medium or on tryptic soy agar (TSA; BD, Rockville, MD, USA) solid medium with 10% newborn bovine serum (Sijiqing, Hangzhou, China) at 37 °C. For TF1 susceptibility tests, the strain was cultured in Mueller–Hinton broth (MHB; BD, Rockville, MD, USA) or on MHB agar (MHA; BD, Rockville, MD, USA) at 37 °C. TF1 (ChemFaces, Wuhan, China; CAS number: 4670-05-7; molecular weight; 564) solution was prepared with the following steps: 163,840 μg of powder were dissolved in a 10 mL constant volume bottle to obtain the standard solution. The solution was then filtered through a 0.22 μm filter. A culture medium was used to dilute the solution to the appropriate concentration according to the experiments, when necessary.

### 4.2. Susceptibility Testing and Time-Killing Curve of SC19 against TF1

MIC and MBC assays were performed according to the Clinical and Laboratory Standards Institute (CLSI) guidelines [40]. Briefly, the bacteria cells were grown in MHB to the exponential phase and the resulting culture was suspended into MHB at a density of 5 × 10^5^ CFU/mL, and a series of dilute concentrations of TF1 (32, 64, 128, 256, 512, 1024, 2048, 4096, and 8192 µg/mL) were added into 96-well plates. The time-killing curve of TF1 against SC19 was conducted, as described previously [41]. The overnight culture was subcultured into a fresh medium with TF1 at MIC and MBC. SC19, cultured without TF1, was used as a control. Growth curves were measured by 600 nm (Victor Nivo, PerkinElmer, Waltham, MA, USA) and CFU counts were taken every hour for 8 h at 37 °C.

### 4.3. Electron Microscope Observation of Bacterial Integrity

The effect of TF1 on the bacterial integrity of SC19 was examined by TEM and SEM, as described previously [42]. The bacteria cells cultured in TSB to the exponential phase were coincubated with TF1 at MIC at 37 °C for 4 h, followed by fixation with 2.5% glutaraldehyde at 4 °C overnight. Thereafter, the bacterial cells were treated with 1% osmium tetroxide for 2 h at room temperature and dehydrated using a series of diluted ethanol. For TEM, the dehydrated cells were embedded in epoxy resin and cell morphology was analyzed by an HT7700 TEM (HITACHI, Tokyo, Japan). For SEM, the dehydrated cells were coated with a 10-nm-thick gold layer for 30 s and observed using a SU8100 SEM (HITACHI, Tokyo, Japan).

### 4.4. Effect of TF1 on Hemolytic Activity of Suilysin

To evaluate TF1′s ability to disturb hemolysis, the hemolytic activity of the SC19 culture was measured as described previously with minor modification [43]. Briefly, the bacteria cells cultured in TSB to the exponential phase were coincubated with TF1 at MIC and MBC at 37 °C for 4 h, and the supernatant was collected after centrifugation for 2 min at 10,000× *g* at 4 °C. The test samples (100 μL) were incubated with a 2% sheep erythrocyte suspension (100 μL) in PBS for 2 h at 37 °C and TSB was used as a negative control. Finally, the resulting culture supernatant was measured at 550 nm using a multifunctional enzyme reader (Victor Nivo, PerkinElmer, Waltham, MA, USA).

To evaluate the effect of TF1 on the hemolytic activity of suilysin in the SC19 culture supernatant, the bacteria cells were cultured in TSB to the postexponential phase. Subsequently, the bacterial culture supernatant was collected after centrifugation for 10 min at 10,000 r at 4 °C. The supernatant was then incubated with TF1 at MIC and MBC for 2 h at 37 °C. Finally, the resulting supernatant incubated with sheep erythrocyte suspension was measured according to the method described above.

### 4.5. Effect of TF1 on Biofilm Formation of SC19

The Biofilm formation assay was conducted as described previously with minor modifications [44]. The bacteria cells in the exponential phase in TSB were cocultured with TF1 at MIC and MBC for 48 h at 37 °C in 48-well microtiter plates. TSB was used as a negative control. The cultures were then discarded, and the plates were washed thrice with PBS. The formed biofilms were stained with 0.1% crystal violet for 30 min and washed thrice with double-distilled H_2_O and then dried in air. Subsequently, 200 µL of 95% ethanol was added, and the plates were measured at an absorbance of 590 nm using a multifunctional enzyme reader (Victor Nivo, PerkinElmer, Waltham, MA, USA).

### 4.6. Lactate Dehydrogenase (LDH) Cytotoxicity Assay of TF1

A lactate dehydrogenase release assay was used to evaluate cell membrane integrity, according to the manufacturer’s protocols (Beyotime, Shanghai, China). In brief, Nptr cells were treated with different concentrations of TF1 (MBC, MIC) for 30 min at 37 °C in 96-well microtiter plates. Untreated Nptr cells were used as a negative control. After incubation, the cell-culture supernatant was mixed with the substrate and incubated for 1 h at 37 °C. The plates were measured at an absorbance of 490 nm and 600 nm using a multifunctional enzyme reader (Victor Nivo, PerkinElmer, Waltham, MA, USA). Finally, the percentage of cytotoxicity was calculated according to the manufacturer’s protocols.

### 4.7. Test the Anti-Adhesion Activity of TF1 on SC19

To evaluate the effect of TF1 on the adhesion of SC19 to the epithelial cell Nptr, we performed an experiment as described previously with minor modifications [45]. Nptr cells were co-incubated with SC19 at exponential phase (MOI = 100:1) and TF1 at different concentrations (MIC, 1/2MIC, and 1/4MIC) for 30 min at 37 °C. The cellular monolayers were washed with PBS thrice and lysed in 1 mL of sterile double-distilled H_2_O. Adherent bacteria were determined by plating a serial dilution of the lysates on TSA agar.

### 4.8. Test the Anti-Infection Activity of TF1 on SC19

All mice used in this study were purchased from the Hubei Experimental Animal Research Center (Wuhan, China). Animal studies were conducted in strict accordance with the animal welfare guidelines of the World Organization for animal health. The animal study was approved by the Ethics Committee of the Institute of Animal Husbandry and Veterinary, Hubei Academy of Agricultural Sciences (Wuhan, China; identification code: XCXK(E)2020-0018), date of approval: 20220520). Thirty six-week-old female specific-pathogen-free (SPF) Kun-Ming mice were used as an infection model to evaluate the protection afforded by TF1. Each group comprised 10 mice. Group 1 was administered TF1 at the dose of 50 mg/kg intragastrically. After 30 min, group 1 and group 2 were intraperitoneally infected with 4 × 10^8^ CFU/mouse of SC19. Group 3 was administered PBS and was applied as a negative control. The morbidity, mortality, and clinical symptoms of the mice in each group were monitored for 7 days.

In addition, another batch of 45 six-week-old female SPF Kun-Ming mice was used to evaluate the effect of TF1 on bacterial load in different organs. Each group comprised 15 mice. Group 1 was administered TF1 at the dose of 50 mg/kg intragastrically. After 30 min, group 1 and group 2 were intraperitoneally infected with 1 × 10^8^ CFU/mouse of SC19. Group 3 was administered PBS and was applied as a negative control. At 1 day, 2 days, and 3 days postinfection (dpi), 5 mice in each group were sacrificed to collect brain, lung, spleen, and blood, which were used for bacteria counts. Moreover, blood was also used to measure TNF-α and IL-6 production.

### 4.9. Molecular Docking Assay

Molecular docking of the interaction mode between TF1 and Sly was conducted by the molecular operating environment (MOE) [46,47]. The chemical structure of TF1 was downloaded from PubChem, and the three-dimensional (3D) structure of Sly was obtained from the Protein Database (PDB).

### 4.10. qRT-PCR to Analyze the Cell Division and Virulent Related Genes at the mRNA Level

To further explore the underlying mechanism of the antibacterial activity of TF1, the RNA expression of cell division and virulent-related genes of SC19 was evaluated. Briefly, SC19 was grown to the mid-log phase. Then, TF1 was added at 512 μg/mL and cocultured at 37 °C for 4 h. Total RNA was reverse transcribed using the HiScript II First-Strand cDNA Synthesis Kit (Vazyme, Nanjing, China) following the recommended protocol. Primers were listed in Table 4. The 16S rRNA gene was chosen as the internal control. Fold change >2 and *p* < 0.05 were used to represent up- or downregulation.

## 5. Conclusions

This study profiled the antibacterial effects of TF1 on *S. suis*, the MIC against SC19 was 512 µg/mL. TF1 exhibited significant inhibitory effects on *S. suis* growth, hemolytic activity, and biofilm formation, and caused extensive damage to *S. suis* cells. The cell model and mouse model infection experiments indicated that TF1 decreased the adhesion ability and virulence of SC19, and reduced the production of IL-6, and TNF-α in infected mice. In addition, the hemolysis test revealed the direct interaction between TF1 and Sly, while molecular docking showed TF1 had a good binding activity with the Glu198, Lys190, Asp111, and Ser374 of Sly. Moreover, TF1 inhibited the expression of Sly genes and other virulence genes. This study suggested that TF1 is a promising phytochemical compound for treating *S. suis* infection.

## Figures and Tables

**Figure 1 ijms-24-07442-f001:**
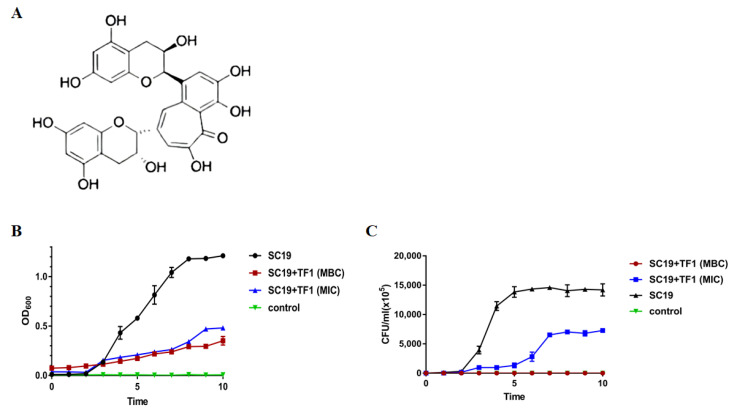
TF1 antibacterial activity. (**A**) Chemical structure of TF1. (**B**) Kinetics of the killing activity of TF1 against SC19 were monitored by OD600nm at the indicated times. (**C**) Kinetics of the killing activity of TF1 against SC19 were monitored by CFU counts at the indicated times.

**Figure 2 ijms-24-07442-f002:**
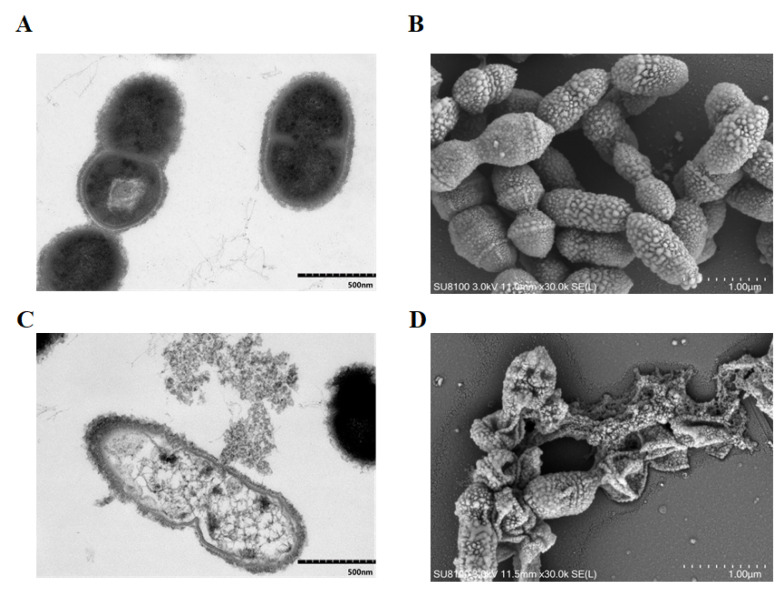
Cellular shape and structure analysis of SC19. (**A**) TEM analysis of untreated SC19, the bar at the bottom right means 500 nm. (**B**) SEM analysis of untreated SC19, the bar at the bottom right means 1 μm. (**C**) TEM analysis of SC19 treated with TF1 at MIC, bacterial cell presented vacuolation degeneration, the bar at the bottom right means 500 nm. (**D**) SEM analysis of SC19 treated with TF1 at MIC, bacterial cell presented shrinkage, cell size reduction and perforation of the cell surface, the bar at the bottom right means 1 μm.

**Figure 3 ijms-24-07442-f003:**
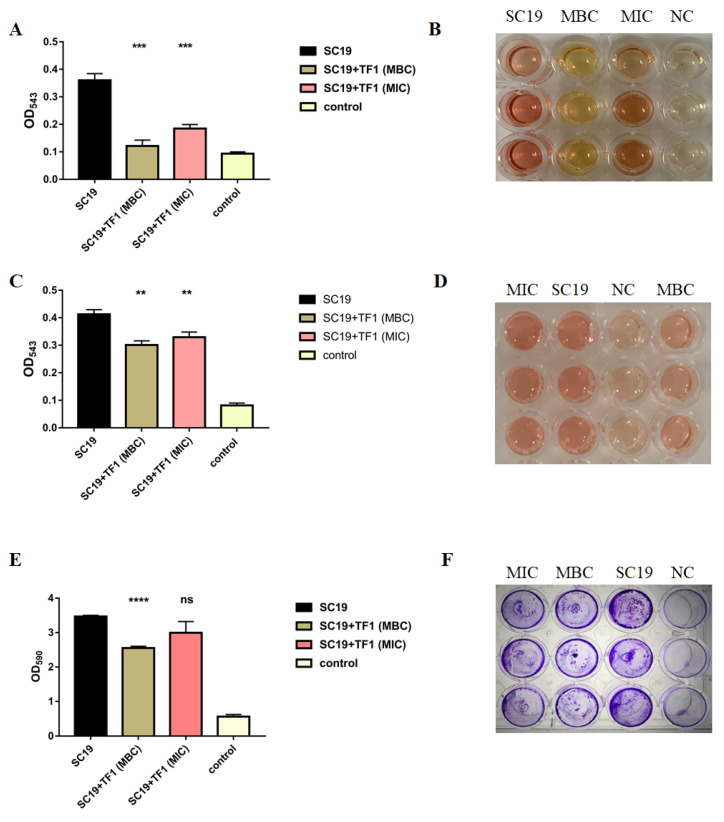
Hemolytic activity and biofilm formation of SC19 affected by TF1. (**A**,**B**) Hemolytic activity analysis of SC19 culture affected by TF1. Absorption was measured at 543 nm to determine Sly activity. TSB was used as a negative control. (**C**,**D**) Direct interaction of TF1 and Sly revealed by hemolytic activity analysis of SC19 culture supernatant treated by TF1. Absorption was measured at 543 nm to determine Sly activity. TSB was used as a negative control. (**E**,**F**) Biofilm formation analysis of SC19 affected by TF1. Absorption was measured at 590 nm to determine biofilm production. TSB was used as a negative control. The height of the bars indicates the mean values for the relative expression data ± SEM (ns, *p* > 0.05; ** *p* < 0.01; *** *p* < 0.001; **** *p* < 0.0001).

**Figure 4 ijms-24-07442-f004:**
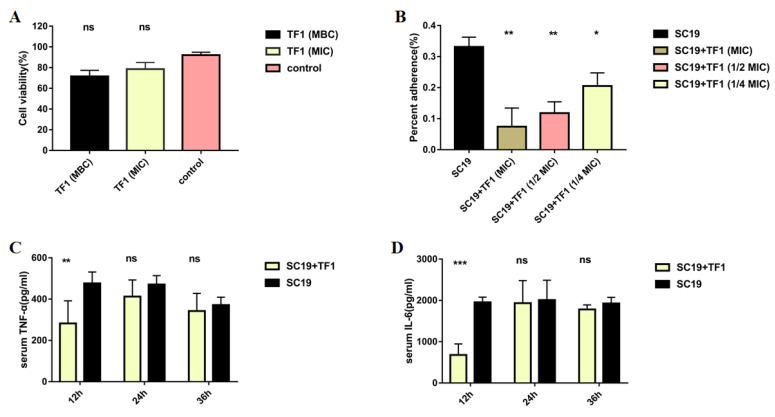
TF1 inhibited the adhesion ability of SC19 and SC19-mediated cytokine production. (**A**) TF1 had no cytotoxic effect on the epithelial cell Nptr through LDH release measurements. (**B**) Adhesion ability analysis of SC19 affected by TF1. (**C**) TNF-αproduction in serum of SC19-infected mice affected by TF1. (**D**) IL-6 production in serum of SC19-infected mice affected by TF1. The height of the bars indicates the mean values for the relative expression data ± SEM (ns, *p* > 0.05; * *p* < 0.05; ** *p* < 0.01; *** *p* < 0.001).

**Figure 5 ijms-24-07442-f005:**
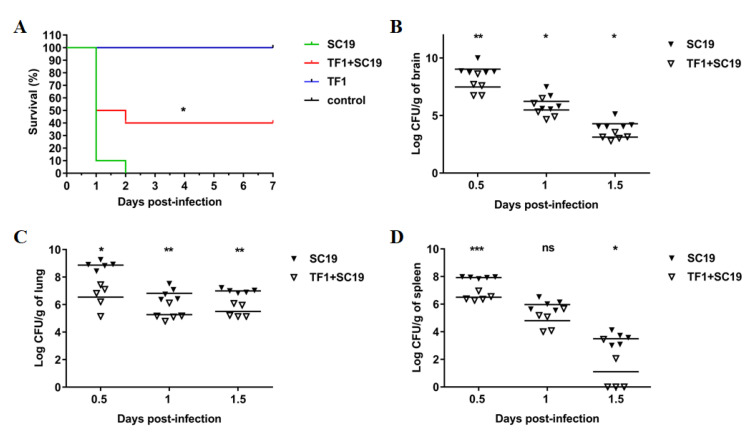
TF1 reduced the pathogenicity of SC19. (**A**) TF1-protected mice against SC19 infection, a significant difference in survival between different groups was analyzed by log rank test. (**B**) Bacterial burdens in the brains of the SC19-infected mice. (**C**) Bacterial burdens in the lungs of the SC19-infected mice. (**D**) Bacterial burdens in the spleens of the SC19-infected mice. The height of the bars indicates the mean values for the relative expression data ± SEM (ns, *p* > 0.05; * *p* < 0.05; ** *p* < 0.01; *** *p* < 0.001).

**Figure 6 ijms-24-07442-f006:**
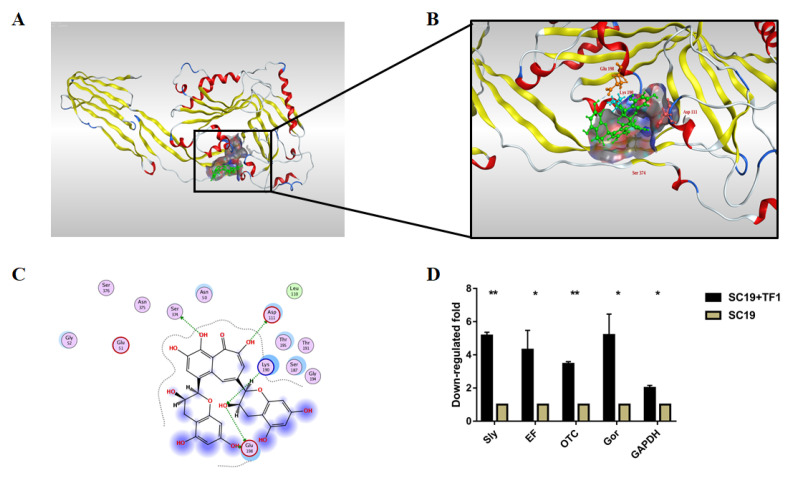
Molecular docking and qRT-PCR results. (**A**) 3D structure of Sly docked with TF1. (**B**) Enlarged view of the pocket after molecular docking of TF1 and Sly protein, where the amino acids Glu198, Lys190, Asp111, and Ser374 react with TF1. (**C**) Interactions between the binding site of TF1 and the amino acid of Sly protein. The hydroxyl groups of TF1 are linked to the amino acids of the Sly protein, the five hydrogen bonds are indicated with the green dotted arrows. (**D**) Virulence-related genes expression analysis of TF1 treated SC19 compared to untreated SC19. The height of the bars indicated the mean values for the relative expression data ± SEM * *p* < 0.05; ** *p* < 0.01).

**Table 1 ijms-24-07442-t001:** Minimum inhibitory concentrations (MICs) and minimum bactericidal concentrations (MBCs) of theaflavin (TF1) against *Escherichia coli* and *Streptococcus suis*.

Bacterial Strain	MIC μg/mL	MBC μg/mL
*Escherichia coli*, K99	2048	4096
*Escherichia coli*, K88	2048	4096
*Escherichia coli*, 987P	2048	4096
*Escherichia coli*, ATCC25922	4096	4096
*Streptococcus suis*, serotype 2	512	2048
*Streptococcus suis*, serotype 7	512	2048
*S.equi subsp. Zooepidimicus*	512	2048

**Table 2 ijms-24-07442-t002:** 32 binding pockets of Sly for TF1 were simulated by MOE.

Site	Size	PLB	Hyd	Side	Residues
1	68	3.37	16	40	1:(ANS50 GLU51 GLY52 ASN82 ASN83 SER84 ASP86 ILE87 GLN107 LEU109 LEU110 ASP111 ASN112 SER187 LYS190 THR191 LYS192 PHE193 GLY194 THR195 ILE372 LEU373 SER374 ASN375 SER376)
2	36	2.20	20	30	1:(TYR54 ILE55 TYR184 SER185 MET186 SER187 PHE208 APS209 VAL211 ASN212 GLU377 TYR378 ILE379 THR381)
3	57	1.40	14	38	1:(MET182 PYR184 VAL211 GLU214 GLU215 LYS216 GLN217 SER281 SER282 ARG283 SER284 THR285 GLN286 VAL287 GLN288 ALA289 GLU380 THR381 THR382 SER383)
4	21	1.16	10	17	1:(THR62 GLU63 LEU65 PHE70 GLU409 VEL410 SER411 TYR412 VEL419 GLU421 ASN445)
5	57	0.96	25	36	1:(ILE97 TYR98 PRO99 ARG147 VAL150 ARG151 VAL154 ASN155 LEU158 TYR228 TYR270 GLY325 GLY354 VAL355 PRO356)
6	11	0.83	8	27	1:(PHE70 VAL72 ARG74 THR384 HIS36 TYR412 GLU418 ARG447)
7	43	0.7616	16	36	1:(ASN82 SER84 ASP86 ILE87 ALA88 ILE90 ASN112 GLM177 ASP179 LYS192 PHE193 6LY194 THR195 SER196 ASN22 L’S24 PHE275)
8	27	0.59	13	23	1:(ARG272 MET274 ILE319 GLY322 ASP323 LYS340 ILE341 GLU344 GLY345 ALA346 TYR348 GLY349)
9	51	0.38	21	37	1:(VAL89 ILE90 ASP91 ALA94 ALA95 ILE97 ASP111 ASN112 ASN113 LYS190 THR195 SER196 GLU198 LYS199 TYR359)
10	22	0.21	10	14	1:(LEU128 ASN129 LEU130 PRO131 GLY132 LEU133 ALA134 ASN135GLY136 ASP137 TRP161)
11	32	0.18	11	32	1:(ILE80 GLU180 THR181 MET182 TYR184GLN188 LYS192 GLN288 TYR378 GLU380)
12	16	0.14	5	20	1:(ARG69 SER281 SER282 ARG283 TYR306 VAL385 ASN387)
13	29	−0.11	6	15	1:(HIS386 ASN387 SER388 SER389 ILE442 PRO443 GLY444 ASN445 ALA446 ARG447 LEU474 VAL475 GLV476)
14	19	−0.12	9	23	1:(GLU66 ASN67 ARG69 VAL71 GLY214 GL0215 LYS216 SER281 ARG283 VAL385)
15	11	−0.14	6	18	1:(TYR98 LEU102 LEU116 ILE117SER118 ILE119 ARG121 PRO145 THR146 ARG147)
16	55	−0.16	19	32	1:(GLY170 ASN171 THR172 GLN173 ALA174 LEU176 PHE223 GLN225 TYR2271LE295 LYS296 GLV297 ILE342 GLU343 ALA345 ARG347)
17	21	−0.19	18	21	1:(THR285GLN286 ALA289 ALA290 ILE300 ALA304 GLU305 TYR306 GLN307 ILE309 LEU310)
18	18	−0.26	9	12	1:(ARG447ASN448 LEU 449 ASP 471 LEU 472 PRO473 LEU 474)
19	28	−0.44	19	23	1:(ACE31 ASP32 ILE33 TYR36 VAL248 GLU249 LEU251 LYS252)
20	26	−0.52	6	19	1:(ASN67 GLY68 ARG69 ASN387SER388 SER389 ALA390 GLN441 GLY476 GLN477 LEU498)
21	9	−0.64	5	9	1:(GLU46 ILE47 LEU48 THR49 ASP106 GLN107 LEU110)
22	17	−0.64	10	20	1:(GLU175 LEU176GLN177 LYS224 ILE226 THR229 SER269 SER358)
23	10	−0.65	9	14	1:(ALA293 ALA294 ILE295 GLY297 ASP299 ILE300 SER301 LYs339 ILE342 GLU343)

**Table 3 ijms-24-07442-t003:** Molecular docking scores.

	Mol	S	rmsd_refine	E_conf	E_place	E_score1	E_refine
1	135403798	−6.3573	1.7533	101.7029	−88.9436	−13.1194	−26.4553
2	135403798	−6.2809	2.1405	101.6749	−124.5373	−12.5693	−21.9376
3	135403798	−6.1305	2.0692	103.7832	−92.5910	−12.0843	−19.3180
4	135403798	−6.1281	2.9734	101.0886	−84.2101	−13.3046	−23.3519
5	135403798	−6.0669	4.2344	102.9650	−85.8538	−15.1033	−20.1235

**Table 4 ijms-24-07442-t004:** Primers used for qRT-PCR in this study.

Primers	Primers Sequence (5′-3′)	Reference or Source
16S rRNA-F	CATCCATAACAGCCATACCAG	[48]
16S rRNA-R	TAAACCACATGCTCCACCGC	
*gapdh*-F	GCTGAAGAAGTAAACGCTGCT	[48]
*gapdh*-R	GTCGCATCAAACAATGAACC	
*sly*-F	AGTCAGTTTGGCACTCGTAGG	This work
*sly*-R	TTGTGGCTCGTAAGTCAAGC	
*ef*-F	TCCAATCACAGATCCAGATAGCG	[48]
*ef*-R	CTGACCCATTTGGACCATCTAAG	
*gor*-F	GTTCACGCGCATCCTACG	[48]
*gor*-R	TACCAGGAATAGCAGGGAC	
*otc*-F	TTGCCCTCTTGAAGCCATACCA	This work
*otc*-R	TTCCATTTCTTCTACGCCGAAT	

## Data Availability

Data are contained within the article.

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
