# Peer review of "Theaflavin Ameliorates Streptococcus suis-Induced Infection In Vitro and In Vivo"

_ijms, 2023, doi:10.3390/ijms24087442_

Round 1
Reviewer 1 Report
In this manuscript, the authors present the experimental data regarding the anti-Streptococcal suis activity of theaflavin. Overall, the authors present interesting data that would help with the development of antibacterial drugs. There are a few specific comments that I'd like for the authors to address:
# In the abstract, the authors mention "no cytotoxicity", this was not developed in the body of the manuscript. Alos, is it "no cytotoxicity" or "low cytotoxicity"?
# In the introduction, it would be helpful to include the current treatment for SS, in pigs and humans and some discussion about the resistance to those antibacterials.
# In the results, it would be useful to include the positive control of the antibacterial activity, especially if the controls are the antibacterial commonly used to treat these infections.
# In the intro, results, and discussion, a better explanation regarding Sly as a target would enhance the manuscript. Is it a validated target? What is the function in the bacterial cell?
# In the discussion, the authors mention TF1 is an effective antagonist for Sly, what is the antagonistic effect? It looks more like an inhibitory effect based on their discussion/results.
# The following are grammar comments/formatting: 1) Under section 2.2. it would be better to better cite/comment on figure 2. 2) In section 2.4., What is LDH? Also, it is never a good idea to start a sentence with an acronym. 3) Section 2.5. seems very disorganized, and difficult to understand and follow. 4)Table 2 seems too long and not completely explained, it probably can be condensed and the authors could include the full table in the supplementary material. 5) In the discussion, the second sentence needs to be revised, it is a very strong statement and the authors forget that there are many toxic natural products as well.
Author Response
# In the abstract, the authors mention "no cytotoxicity", this was not developed in the body of the manuscript. Alos, is it "no cytotoxicity" or "low cytotoxicity"?
RE: Thank you for your nice suggestion. In the Materials and Methods section, Lactate dehydrogenase (LDH) cytotoxicity assay of TF1 was performed. LDH Assay Kit provides a sensitive, quick, and easy way to detect LDH released from damaged cells. In the LDH assay, LDH converts lactate to pyruvate and NADH, which reduces a proprietary probe to an intensely fluorescent product. The amount of fluorescence is directly proportional to the number of damaged cell. So, LDH release measurements were performed to determine whether TF1 could be cytotoxic to epithelial cell Nptr. Figure 4A showed that TF1 with different concentration did not injure Nptr at all.
# In the introduction, it would be helpful to include the current treatment for SS, in pigs and humans and some discussion about the resistance to those antibacterials.
RE: Thank you for your nice suggestion. We have added information about the current treatment for SS, in pigs and humans and some discussion about the resistance to those antibacterials in the revised manuscript.
# In the results, it would be useful to include the positive control of the antibacterial activity, especially if the controls are the antibacterial commonly used to treat these infections.
RE: Thank you for your nice suggestion. It is helpful of your suggestion, in view of animal welfare, we did not set the positive control of antibacterial group, and to maintain consistent throughout this study, we did not set the positive control of antibacterial group for other experiment. But, the positive and negative controls were designed throughout this study, so the data can be trusted without doubt.
# In the intro, results, and discussion, a better explanation regarding Sly as a target would enhance the manuscript. Is it a validated target? What is the function in the bacterial cell?
RE: Thank you for your nice suggestion. SLY, a secreted protein, can creates pores in the target cells membranes. Sly is critical virulence factor for S. suis in the successful colonization of host cells and immune system evasion by the host. S. suis strains with high levels of Sly production are more likely to cause high mortality in infected mice than non-virulent strains, thus indicating that the pathogenicity of S. suis can be attenuated by lowering the production of Sly.
Based on above information, researchers find inhibitors of Sly , such as Baicalein, Amentoflavone, Ellipticine Hydrochloride, which can inhibit the hemolytic activity of Sly by directly binding to Sly and improved the survival rate of mice infected with S. suis (Wang C, Lu H, Liu M, et al. Effective Antibacterial and Antihemolysin Activities of Ellipticine Hydrochloride against Streptococcus suis in a Mouse Model )( Lu H, Li X, Wang G, et al. Baicalein Ameliorates Streptococcus suis-Induced Infection In Vitro and In Vivo. Int J Mol Sci. 2021;22(11):5829.) (Shen X, Niu X, Li G, Deng X, Wang J. Amentoflavone Ameliorates Streptococcus suis-Induced Infection In Vitro and In Vivo. Appl Environ Microbiol. 2018;84(24):e01804-18.).
Therefore, Sly is a unique molecular target of the novel drugs against S. suis infection. We have added more information about Sly in the revised manuscript.
# In the discussion, the authors mention TF1 is an effective antagonist for Sly, what is the antagonistic effect? It looks more like an inhibitory effect based on their discussion/results.
RE: Thank you for your nice suggestion. We have changed antagonist into inhibitor in the Discussion in the revised manuscript.
# The following are grammar comments/formatting: 1) Under section 2.2. it would be better to better cite/comment on figure 2. 2) In section 2.4., What is LDH? Also, it is never a good idea to start a sentence with an acronym. 3) Section 2.5. seems very disorganized, and difficult to understand and follow. 4)Table 2 seems too long and not completely explained, it probably can be condensed and the authors could include the full table in the supplementary material. 5) In the discussion, the second sentence needs to be revised, it is a very strong statement and the authors forget that there are many toxic natural products as well.
RE: Thank you for your nice suggestion. 1) figure 2. 2 have been better cited in the revised manuscript. 2) Lactate dehydrogenase (LDH) Assay Kit provides a sensitive, quick, and easy way to detect LDH released from damaged cells, and we have changed LDH into Lactate dehydrogenase in the revised manuscript. 3) Section 2.5. has been reorganized in the revised manuscript. 4) Table 2 have been transferred to supplementary material Table S1 in the revised manuscript. 5) the second sentence in the disccusion has been revised in the new version.
Reviewer 2 Report
The article by Ting Gao et al. describes the effect of Theaflavin on S. suis.
The paper describes interesting results, but struggles to describe the interest of this work (questionable antibiotic resistance on S. suis, lower virulence than S. pneumoniae for example). It is therefore necessary to rework the introduction to justify its publication. Moreover, are the results observed transposable to a veterinary and/or human health use (in terms of administered concentration and therapeutic margin)?
In addition, some problems need to be corrected:
- put in italics; the names of bacteria, in vitro, in vivo.
- Figure 1 : A : not interesting because not readable.
- Figure 1 C and D : time in hours?
- Figure 3 E : the standard deviation is important for SC19TF1 MIC compared to the others.
- Figure 6 D : isn't the observed decrease due to the decrease of bacterial concentration?
- methods : justify the quantity of powder diluted in 1 ml
- correct the fonts and prefer passive forms of words.
Author Response
The paper describes interesting results, but struggles to describe the interest of this work (questionable antibiotic resistance on S. suis, lower virulence than S. pneumoniae for example). It is therefore necessary to rework the introduction to justify its publication. Moreover, are the results observed transposable to a veterinary and/or human health use (in terms of administered concentration and therapeutic margin)?
RE: Thank you for your nice suggestion. The introduction has been reworked in the revised manuscript. “are the results observed transposable to a veterinary and/or human health use” is a very great question. It takes a long time for laboratory research to lead to clinical use. We screened and evaluated TF1 as a candidate compound for pharmacological activity in vitro (e.g. at the cellular level) and in vivo (e.g. at the animal level). Based on the experimental data, TF1 could be promoted to "lead compound". The "lead compound" has the potential for subsequent development as a "drug candidate". Clinical researches are the continuous exploration and verification process of drug therapeutic effect, drug dosage and adverse reactions. To find out administered concentration and therapeutic margin of human or animal, large scale clinical trials are needed before any conclusions could be made.
- put in italics; the names of bacteria, in vitro, in vivo.
RE: Thank you for your nice suggestion. The names of bacteria, in vitro, in vivo have been in italics in the revised manuscript.
- Figure 1 : A : not interesting because not readable.
RE: Thank you for your nice suggestion. Figure 1 : A has been deleted in the revised manuscript.
- Figure 1 C and D : time in hours?
RE: Thank you for your nice suggestion. Since Figure 1 : A has been deleted, Figure 1 C and D become Figure 1 B and C. Yes, growth curves were indicated in hours.
- Figure 3 E : the standard deviation is important for SC19 TF1 MIC compared to the others.
RE: Thank you for your nice suggestion. Biofilm formation analysis of SC19 affected by TF1 was repeated many times, but it had no effect at MIC.
- Figure 6 D : isn't the observed decrease due to the decrease of bacterial concentration?
RE: Thank you for your nice suggestion. No, we set internal reference in qRT-PCR, the 16S rRNA gene was chosen as internal control. The inaccuracy of final product quantification can be eliminated by adding internal reference. In addition, SC19 was grown to the mid-log phase. Then, TF1 was added at 512 μg/ml and co-cultured at 37 °C for 4 h, the bacterial concentration of control group and treated group were the same. So, the observed decrease was not due to the decrease of bacterial concentration.
- methods : justify the quantity of powder diluted in 1 ml
RE: Thank you for your nice suggestion. 163840 μg of powder was dissolved in 10 mL constant volume bottle to obtain the standard solution, and we have added this sentence in the revised manuscript.
- correct the fonts and prefer passive forms of words
RE: Thank you for your nice suggestion. We have corrected the fonts and prefer passive forms of words in the revised manuscript.
Round 2
Reviewer 1 Report
The authors have addressed all my concerns. The manuscript is acceptable for publication in my opinion.
Reviewer 2 Report
The manuscript has been revised according to my previous comments and is suitable for publication to my opinion.